# Potassium Lactate as a Strategy for Sodium Content Reduction without Compromising Salt-Associated Antimicrobial Activity in Salami

**DOI:** 10.3390/foods10010114

**Published:** 2021-01-07

**Authors:** Francis Muchaamba, Helena Stoffers, Ralf Blase, Ueli von Ah, Taurai Tasara

**Affiliations:** 1Institute for Food Safety and Hygiene, Vetsuisse Faculty, University of Zürich, 8057 Zürich, Switzerland; tasarat@fsafety.uzh.ch; 2Agroscope, 3003 Bern, Switzerland; helena.stoffers@agroscope.admin.ch (H.S.); ralf.blase@agroscope.admin.ch (R.B.); ueli.vonah@agroscope.admin.ch (U.v.A.)

**Keywords:** potassium lactate, sodium chloride, salami, sausage, antimicrobial, *Listeria innocua*, *Listeria monocytogenes*

## Abstract

Reformulating recipes of ready-to-eat meat products such as salami to reduce salt content can mitigate the negative health impacts of a high salt diet. We evaluated the potential of potassium lactate (KL) as a sodium chloride (NaCl) replacer during salami production. NaCl and KL stress tolerance comparisons showed that four food-derived *Listeria innocua* isolates were suitable as biologically safe *Listeria monocytogenes* surrogates. Effects of the high salt (4% NaCl) concentration applied in standard salami recipes and a low salt (2.8% NaCl) plus KL (1.6%) combination on product characteristics and growth of contaminating *Listeria* and starter culture were compared. Simulated salami-ripening conditions applied in meat simulation broth and beef showed that the low salt plus KL combination retained similar to superior anti-*Listeria* activity compared to the high salt concentration treatment. Salami challenge tests showed that the low NaCl plus KL combination had comparable anti-*Listeria* activity as the high NaCl concentration during ripening and storage. No significant differences were detected in starter culture growth profiles and product characteristics between the high NaCl and low NaCl plus KL combination treated salami. In conclusion, KL replacement enabled a 30% NaCl reduction without compromising the product quality and antimicrobial benefits of high NaCl concentration inclusion.

## 1. Introduction

*Listeria monocytogenes* is a bacterial contaminant in ready-to-eat (RTE) foods resulting in serious foodborne illnesses and economic losses to the food industry [1,2,3]. The risk of illness or death per portion among RTE foods contaminated by *L. monocytogenes* is high in deli meats like bologna type sausage, salami, and smoked salmon in comparison to other foods [2,3,4]. Monitoring data have shown that *L. monocytogenes* is isolated at different frequencies from meat and meat products [2,3,4]. An association between contaminated RTE meat in outbreaks and sporadic listeriosis infections is well documented [5], for example, the biggest listeriosis outbreak recorded to date was linked to a RTE bologna type sausage product [6]. As a result, hurdle technology including several bactericidal and bacteriostatic intervention strategies against these pathogens have been developed to improve food safety, though with varying success levels [7,8,9]. Among these interventions, low water activity and osmotic stress induced through the addition of salts like sodium chloride (NaCl) are used. Also, protective cultures and antimicrobials such as nisin, organic acids and their salts, such as diacetate and lactate are applied [7,9,10,11,12]. Most of these have bacteriostatic activity prolonging food shelf life by inhibiting the growth of spoilage and pathogenic bacteria during handling and storage under ideal refrigeration conditions and even under abusive temperature conditions [7,12,13]. 

*L. monocytogenes* employs a plethora of mechanisms to survive harsh environmental conditions encountered in foods and food processing plants resulting in resistance to commonly used food safety and preservation interventions [7,14,15,16,17]. In case of osmotic stress tolerance, these mechanisms include but are not limited to alternative sigma factors and cold-shock proteins as well as protein systems involved in the accumulation of osmo-protective solutes [17,18,19]. Moreover, some of these interventions have the drawback of priming the bacterium for intracellular life thereby increasing its virulence. For example, low pH, NaCl, and cold stress prime for resistance to gastric acid stress [14,20,21,22]. Additionally, some measures induce cross-protection to other hurdle procedures as is observed with salt and cold stress [14,15,23,24]. *L. monocytogenes* possesses both intrinsic and acquired abilities to survive, adapt, and grow in the presence of disinfectants, antimicrobial peptides, high osmotic pressure, low pH, and refrigeration temperature. This has necessitated the continued search for novel interventions or combinations of already existing measures for improved control of this bacterium in food and processing environments [17,19,25]. 

The control of *L. monocytogenes* in food processing facilities has remained problematic and costly mainly due to the wide distribution of this bacterium which makes introduction into a facility relatively easy [5,17,26,27]. Once introduced eradicating it is even more challenging as the bacterium has evolved several systems that it can deploy such as biofilm formation and disinfectant tolerance which allows for its persistence eventually contaminating foods and causing outbreaks [5,27,28,29,30,31].

Meanwhile, some of the measures to control the growth of bacteria in food such as the addition of nitrates and high NaCl to food have been linked to various negative health effects. Disproportionate sodium (>2 g/day) and potassium intake (<3.5 g/day) has been linked to hypertension and increased cardiovascular-related diseases and death [13,32,33,34,35]. Cardiovascular diseases have high direct and indirect costs on economies all over the world, for the European Union (EU) and the United States of America the cost is nearing a trillion US dollars per year [13,36,37]. We acquire sodium from sources such as sodium lactate, sodium glutamate, and meat, however, the principal source of dietary sodium is salt. Daily salt or sodium intake for most populations is estimated to be higher than the recommended daily intake quantities [13,35,38]. Salt intake of less than 5 g a day for adults helps to moderate blood pressure and lower the risk of cardiovascular disease, stroke, and coronary heart attack [39]. Reducing sodium intake is considered an important public health strategy, with the principal benefit of a corresponding reduction in high blood pressure. It is projected that a reduction of global sodium intake to the recommended daily amounts could result in the avoidance of about 2.5 million mortalities each year [35]. 

Processed foods such as cheese, RTE meals, meat products such as salami and bacon are some of the major dietary sources of sodium because they are high in salt [34]. While foods such as bread and processed cereal which do not particularly have high salt content contribute to salt intake because they are consumed frequently in large amounts [13,35,40]. Lifestyles are changing eating patterns to favor consumption of processed foods and RTE foods most of which usually have high salt levels in part for flavor and as a preservative [13,35]. At the same time, people are eating less potassium-rich foods such as fruits and vegetables [35,41]. These foods are key components of a healthy diet and contain potassium, which contributes to blood pressure reduction. 

Meat and processed meat products comprise one of the major sources of sodium in most populations, contributing to an estimated 21% of daily sodium intake [13]. To mitigate the health impacts of a high salt diet some producers are reformulating or inventing recipes to reduce the salt content of their products [42,43,44]. Reduction of salt concentration in food and finding suitable replacements that maintain functional, flavor, and antimicrobial effects of high salt concentrations is of paramount importance. Previous work using KL alone or in combination with substances such as potassium chloride (KCl) and sodium diacetate on cold smoked salmon, frankfurters, ham, hotdog, fresh and fermented sausage model systems demonstrated that KL or its combinations have antimicrobial activity against *L. monocytogenes* [12,43,45,46,47,48]. These observations highlight KLs’ potential use as a salt replacer under certain conditions albeit with varied success. 

Amongst processed meat products, cured meat products and sausages including salami have the highest levels of sodium, usually above the recommended targets. Therefore, reducing sodium levels in such foods might have a profound effect on overall daily sodium intake. In this vein, a salt reduction study with salami was previously undertaken, establishing that most participants could not detect the difference in quality and flavor between the standard salami recipe with reference salt concentration and salami reduced in NaCl concentration supplemented with potassium lactate (KL), making this a potential good salt replacer in salami production (Agroscope Switzerland unpublished). Furthermore, previous studies have shown that cold and salt stress conditions both of which exist in salami can induce cross-protective systems in *L. monocytogenes* [17]. We, therefore, set out to evaluate the effect of reduction and replacement of NaCl by KL on the antimicrobial effects of the salami recipe against *L. monocytogenes* and its surrogate *L. innocua* strains in artificially inoculated salami model systems.

## 2. Materials and Methods 

### 2.1. Materials

Boneless beef, pork, and pork back fat sourced locally from the same supplier were used for all experiments requiring meat. Non-meat ingredients for the beef trial and salami preparation included salt, raw sausage premix containing nitrate (Scheid-Bonafirm, Scheid-Rusal AG, Gisikon, Switzerland), garlic (Garlic granulate Knospe, Omya Food, Balsthal, Switzerland), pepper (Ground pepper Knospe, Omya Food, Balsthal, Switzerland), and potassium lactate (PURASAL^®^ HiPure P 78%, Purac Bioquimica, Barcelona, Spain). Non-edible casings were used to stuff the mixed minced meat (Artificial casing /fibrous cellulose casings; R2L-D, Kal.50/40 cm, Naturin Viscofan GmbH, Weinheim, Germany).

### 2.2. Bacterial Strains and Culture Conditions

The strains used in this study are listed in Table 1. *L. monocytogenes* strains used were selected because of their public health significance, representing clinical and food-relevant genetic backgrounds. Bacteria were stored at −80 °C as glycerol (20%)-brain heart infusion medium (BHI, Oxoid, UK) cryo-stocks. Using these cryo-stocks, strains were grown overnight on blood agar plates at 37 °C to get single colonies. These colonies were then pre-cultured twice in 10 mL BHI broth (37 °C, 150 rpm) for 16 h, to get stationary phase secondary pre-cultures that were used in all experiments unless otherwise stated. All inoculum used in the study were diluted and standardized to the same optical density at 600 nm (OD_600_), with OD 1 corresponding to 10^9^ colony forming units (CFU) per mL. Post dilution the starting inoculum CFU/mL were confirmed via serial dilutions and plate counts on BHI plates. For OD_600_ based growth assays at 37 °C, OD 1 standardized stationary cultures prepared from each strain were successively diluted (1:100) in PBS then (1:100) in 10 mL BHI to a final concentration of 10^5^ CFU/mL. For all growth assays done at temperatures other than 37 °C, the secondary cultures were grown at 30 °C (150 rpm) prior to being diluted to the specific level used in that experiment. 

### 2.3. Comparison of L. innocua and L. monocytogenes Growth Phenotypes under NaCl and KL Stress

Osmotic stress tolerance was evaluated by assessing *Listeria* growth rate in BHI supplemented with NaCl or KL. NaCl was applied at 4 to 6% (*w*/*v*) representing a commonly used range of normal to high salt concentration in salami production. Secondary pre-cultures of each strain were diluted in BHI to 10^5^ CFU/mL. To assess growth patterns under NaCl, KL, or a combination of the two stresses, 100 μL of BHI supplemented with 0, 8, and 12% NaCl, as well as 5.6% NaCl plus 3.2% (*v*/*v*) KL, were added in triplicate to wells of a 96-well microplate. The wells were already prefilled with 100 μL (10^5^ CFU/mL) of the different *L. monocytogenes* and *L. innocua* strains, resulting in final salt concentrations of 0, 4, and 6% NaCl, and 2.8% NaCl plus 1.6% KL. Cultures were incubated, continually shaking at 25 °C (48 h) or 37 °C (24 h), measuring OD at 600 nm every 30 min in a microplate OD reader (Synergy HT, BioTek Instruments, GmbH, Luzern, Switzerland).

#### Estimation of Growth Curve Parameters

Growth parameter for each strain in BHI only, or BHI supplemented with NaCl or the NaCl plus KL combination were determined based on the growth curves generated from OD_600_ measurements. OD_600_ measurements were transformed to viable count data through the use of calibration curves established inhouse. Briefly, OD_600_ and viable counts used to build these calibrator curves were determined from the same sample at several time points under normal conditions and stress at 37 °C as previously described [55,56]. Using these calibrator curves the OD_600_ growth curve data of each replicate were transformed to CFU/mL then log CFU/mL at each time point [55], these values were then analyzed using DMFit (available from: www.combase.com) as previously described [55,57]. This approach yielded three growth curve parameters: lag phase duration (lambda), growth rate (µ), and final cell density. Using the raw OD_600_ data, area under the curve (AUC), was analyzed using GraphPad Prism. AUC has previously been recommended for use in summarizing growth curves [58,59,60] and was included in the analysis as it captures the effects of stress and growth conditions on the other three parameters described above. The growth parameters for each strain under normal or stress conditions were further analyzed using the R package “*opm*” as previously described [59,61,62]. To facilitate the comparison of the effects of NaCl or NaCl plus KL combination stress on the strains, parameters were normalized relative to their growth under normal conditions vs. growth under stress to correct for inherent variation in growth between strains even in the absence of stress. 

### 2.4. Inoculum Preparation for Simulation Trials and Challenge Tests

For the simulation trials and challenge tests, the EURL *L. monocytogenes* technical guideline principles were followed with modification [63]. Specifically, secondary stationary phase stage BHI pre-cultures of four *L. innocua* strains standardized to OD 1 (10^9^ CFU/mL) were mixed (2.5 mL each strain) to make a strain cocktail that was used in the simulation trials and challenge tests. These cultures were not preadapted to cold growth but were grown at 30 °C because the bacteria were introduced during raw material mixing for simulation trials or ripening which started at 24 °C. The inoculum bacteria numbers were confirmed by enumeration on BHI plates. Furthermore, the final *Listeria* CFU counts introduced by the inoculum in either the MSB, beef, or salami was confirmed on day zero of each experiment. For this, viable cell counts were performed on two samples (2 mL broth and 20 g for minced beef and salami) per batch. These were 10-fold diluted with PBS, homogenized (300 s in a blender bag for beef and salami), serially diluted, plated onto *Listeria* Ottavani and Agosti (ALOA) agar, and incubated for 48 h at 37 ^o^C.

### 2.5. L. innocua Growth Evaluation under Simulated Salami Ripening Conditions in Meat Simulation Broth and on Beef

Trials were conducted on beef and in meat simulation broth (MSB) under simulated salami ripening conditions. Appendix A details the composition of the meat simulation broth. To assess *Listeria* growth in beef and MSB in the presence of SA1 starter culture (Agroscope, Bern, Switzerland), 200 μL of the *L. innocua* strain cocktail was added in triplicate to 200 g minced beef or 200 mL MSB that were supplemented with NaCl alone (4% or 2.8%) or its combination with KL (2.8% NaCl plus 1.6% KL). The SA1 starter culture (*Lactobacillus plantarum* and *Staphylococcus xylosus*) was applied in accordance with the manufacturer’s recommendations. In the meat-based trial, the minced beef, inoculum, starter culture and the different NaCl and KL concentrations and combination were mixed and homogenized by hand massage in a blender bag. The samples were incubated for 70 h (MSB) and 120 h (beef) at 23 °C non-shaking, with the temperature being decreased by 1 °C every 24 h (MSB) and 48 h (beef). Additionally, pH was measured every 30 min using an iCinac pH meter (ASM Alliance, Charles de Gaulle, France) during the incubation period in a water bath (Lauda Ecoline Staredition RE 310, Lauda-Königshofen, Germany).

Viable cell counting was performed on samples (20 g minced beef and 2 mL MSB) collected daily to determine bacterial growth kinetics for each preparation. Minced beef samples were placed in a sterile stomacher bag containing prewarmed PBS to make a 10-fold dilution and homogenized for 300 s in a stomacher. The beef homogenates and MSB samples were 10-fold serially diluted with PBS, spiral plated onto selective agar and incubated as described below. ALOA agar was used for *Listeria* spp., Mannitol salt agar for *Staphylococcus* spp., and DeMan-Rogosa-Sharpe agar for *Lactobacillus* spp. Incubation was at 37 °C for 48 h for colony enumeration of *Listeria* spp. and *Staphylococcus* spp., while for *Lactobacillus* spp. incubation was done anaerobically at 30 °C for 72 h. Viable bacterial cell counts were expressed in CFU/mL of original broth or CFU/g depending on the experiment (limit of detection: 1 CFU/mL; limit of quantitation: 25–250 CFU/plate).

Bacterial chaining levels were evaluated using a Quantom Tx Microbial Cell Counter (Bucher Biotec AG, Basel, Switzerland), following manufacturers recommendations. All samplings were done in triplicate.

#### Comparison of *L. monocytogenes* and *L. innocua* Growth Kinetics in Meat Simulation Broth

To evaluate the potential effects of our potassium lactate-based intervention on *L. innocua* and *L. monocytogenes,* growth kinetics of the two species were compared under simulated salami ripening conditions. Strain cocktails of *L. innocua* (*Li*20869, *Li*20870, *Li*20871, and *Li*20872), *L. monocytogenes* genetic lineage I (N2306, LL195, and N16-0044), and *L. monocytogenes* lineage II (EGDe, N1546, Lm3136, and Lm3163) strains were grown in MSB. The preparation, MSB inoculation, incubation, and growth monitoring of the *Listeria* strain cocktails were performed as described above. Strains used in *L. monocytogenes* strain cocktails comprised isolates of public health significance covering common food and clinical illness-associated genetic backgrounds (Table 1). 

### 2.6. Production, Ripening and Storage of Salami

Salami containing varying amounts of NaCl alone and in combination with KL was produced (Table 2). To achieve this, pork, beef, and pork back fat frozen at −20 °C for 3 days were thawed at 4 °C and ground by passing through a 6 mm plate in a meat grinder (Primus MEW 713-H82, MADO GmbH, Dornhan, Germany). The minced meat and fat were mixed with non-meat ingredients and varying levels of NaCl and KL combinations (Table 2, Appendix A) to form a homogenous mixture in a bowl mixer (Varimixer RN20 VL-2, Rotor Lips AG, Uetendorf, Switzerland). During homogenization, the mixtures were inoculated with Agroscope’s starter culture SA1 at the manufacturer’s recommended rate and a cocktail of *L. innocua* 20869, 20870, 20871, and 20872 strains at a rate of 10^9^ CFU/kg (total inoculum volume 1 mL/kg). The mixture was transferred to a sausage stuffer (Talsa Kolbenfüller H15PA, Talsabell sa, Xirivella, Spain) with non-edible cellulose casings slid along the horn of the stuffer. The meat mixture was stuffed, pinched, and twisted into approximately 200 g salami. The salami batches were subsequently dipped in a solution containing surface culture *Penicillium nalgiovensis* molds (Salami Schimmel weiss, Scheid-Rusal AG, Gisikon, Switzerland) for 10 s and incubated 20 days in a temperature and humidity controlled salami ripening chamber (self-construction by Agroscope, based on common gastro-refrigerator Electrolux, Kälte-Technik, Zollikofen, Switzerland) following Agroscope’s salami ripening protocol recommendations (Appendix A). Post ripening the salami batches were stored at 4 °C for 94 days.

#### 2.6.1. Bacterial Enumeration

To assess bacterial growth during ripening and storage of salami sausages, viable cell counting was conducted periodically. Two 20 g samples per salami recipe were collected and evaluated on days 0, 1, 2, 4, 7, 9, 13, 16, and 20 of ripening. Additionally, to evaluate the bacteria levels after short to long term storage samples were taken at day 27, 64, and 114 representing 7, 44, and 94 days of post ripening storage, respectively. Samples were placed into a sterile stomacher bag to which prewarmed PBS was added to make a 10-fold dilution and homogenized for 300 s in a stomacher. The homogenates were 10-fold serially diluted with PBS, spiral plated onto selective agar and incubated as described above in Section 2.4. The level of bacterial chaining in the samples was also evaluated as previously described. 

#### 2.6.2. Moisture Loss Analyses, pH, Water Activity, and Warner Bratzler Measurements of Salami Samples

Moisture loss was determined by weighing seven salami sausages per recipe at each sampling point on 0, 6, 14, and 20 days of salami ripening and comparing this weight to the initial weight. Moisture loss was used as a proxy for water activity reduction, targeting at least 35% loss. To evaluate if water activity ≤ 0.92 was reached and maintained, at the end of ripening (Day 20) and during storage (Day 114: 94 days of storage at 4 °C) water activity was determined in triplicates according to manufacturer’s recommendations using an AQUALAB water activity meter (Decagon Devices Inc, Washington, DC, USA). For pH determination, a probe was inserted into the core of the salami and pH values were recorded using a pH meter (ebro PHT 830, ebro Electronic GmbH, Urdorf, Switzerland) on 0, 1, 2, 3, 6, 7, 14, and 20 days of salami ripening. The firmness or tenderness of the salami was measured using a Warner–Bratzler testing device following manufactures recommendations (Zwick Universal Testing Machine, BZ2.5/TN1S, ZwickRoell GmbH, Ulm, Germany).

#### 2.6.3. Proximate Composition Analysis

To get an objective assessment of the quality of the salami, proximate composition analysis of the salami was done. The evaluation was performed on three representative samples per each salami batch by a commercial testing laboratory (Swiss Quality Testing Services (SQTS), Dietikon, Switzerland). Specifically, dry matter, final protein, fat, carbohydrate, salt, sodium, ash, and water content of the different salami preparations were evaluated following internationally and locally recognized standards (ISO/IEC 17025:2017 and SN EN ISO/IEC 17025:2018).

### 2.7. Statistical Analysis

All presented data were derived from at least three biological experimental repeats unless stated otherwise. For the salami trial, each sampling time point and treatment, growth inhibition factors were determined. These were calculated as the difference between the mean log CFU count of the previous sampling point and the mean log CFU count of the salami on that particular day divided by the initial time point mean log CFU. Relative log CFU counts and growth potentials were also determined relative to the day zero inoculum or to the control treatments. Statistical analysis of data comparing strains or treatments was done using GraphPad Prism (Version 8.3.0 (328), GraphPad Software, San Diego, CA, USA). *p* values less than 0.05 were considered to be statistically significant based on one-way ANOVA with post-hoc Tukey HSD tests.

## 3. Results and Discussion 

### 3.1. A Low NaCl (2.8%) and KL (1.6%) Combination Displays Superior Antimicrobial Activity against L. monocytogenes than the High NaCl (4%) Applied Alone in BHI

We initially examined whether a low salt (2.8% NaCl) and KL (1.6%) combination that contains 30% less NaCl than the normal high salt levels (4% NaCl) applied in standard salami production recipes was able to retain desired antimicrobial activity levels as high salt concentrations (4% and 6% NaCl) against *L. monocytogenes*. The selected 1.6% KL inclusion level maintains lactate levels below the 4.8% maximum permissible for meat formulations [64]. Growth assays conducted in BHI broth showed that the low NaCl (2.8%) and KL (1.6%) combination induced antimicrobial activity levels that were similar and, in some instances, even superior to those observed with high (4% and 6%) NaCl concentrations normally applied in salami production (Figure 1, Table 3, Appendix A). Despite a less pronounced impact on lag phase the low NaCl-KL combination was more potent in reducing *L. monocytogenes* growth rate than 4% NaCl resulting in growth rate reduction levels that were similar to those observed at 6% NaCl (Figure 1A,B and Table 3). A temperature-dependent synergistic effect was observed on the antimicrobial activity of NaCl applied alone as well as for the low NaCl plus KL combination since greater *Listeria* growth inhibition was observed at 25 °C than 37 °C (Figure 1C,D). 

Overall, our results showed that KL can be an effective NaCl replacer that does not compromise NaCl-associated antimicrobial effects. The antimicrobial effects provided by KL are postulated to be due to its action as a bacteriostat that extends the lag phase or reduce the growth rate of bacteria, which inevitably prolongs the shelf life of the food products [9,12,45,48,65,66,67]. Furthermore, KL also works through reduction of water activity in food products and causing intracellular acidification of the bacterium [67,68,69,70]. 

### 3.2. Assessing L. innocua Strains as Biologically Adequate Surrogates for L. monocytogenes 

The analysis of non-pathogenic *L. innocua* contributes to an enhanced appreciation of the behavior of its pathogenic close relative *L. monocytogenes* in environments beyond the biosafety level 2 laboratory. The niche co-habitation, genomic synteny, and physiological similarity of the two species support the use of *L. innocua* for predicting *L. monocytogenes* behavior in food and their processing environments [54,71]. However, in some circumstances, it may not be appropriate without prior verification of their analogous phenotypes, as several studies have revealed that the two species may differ in some stress responses [71,72]. Four *L. innocua* strains isolated from food were thus examined for their suitability as biologically safe *L. monocytogenes* surrogates under our study conditions that met the biosafety level criteria necessary for the salami production lab subsequently used in conducting the pretrial and salami ripening experiments described below. 

Growth phenotypes determined in BHI supplemented with 4% NaCl or the 2.8% NaCl plus 1.6% KL combination revealed similar growth response profiles between representative *L. monocytogenes* (EGDe and LL195) and four *L. innocua* (20869, 20870, 20871, and 20872) surrogate strains (Figure 2, Appendix A). The four *L. innocua* strains examined were more stress tolerant than the *L. monocytogenes* reference strain EGDe whereas one strain (*Li*20869) was even more stress tolerant than the listeriosis outbreak *L. monocytogenes* strain LL195 (Figure 2), which has previously been determined to have high osmo-tolerance [73]. Based on this analysis the *L. innocua* strains 20869, 20870, 20871, and 20872 were established as biologically safe adequate *L. monocytogenes* surrogates that could be used to assess the predicted stress survival and growth responses of *Listeria* exposed to 4% NaCl as well as the 2.8% NaCl plus 1.6% KL stress combination. 

### 3.3. Salami Ripening Simulation Trials in Meat Simulation Broth and Beef-Based Models 

Salami ripening simulation trials were conducted in meat simulation broth (MSB) and beef-based models that were artificially inoculated with *Listeria*. The 2.8% NaCl plus 1.6% KL combination showed an improved antimicrobial activity against *Listeria* when compared to the high salt (4% NaCl) concentration during the simulated salami ripening period over 70 h in MSB and 120 h in beef (Figure 3). In both models the low NaCl salt plus KL combination prevented the growth and caused reduction of the original inoculum whereas *Listeria* was still able to grow during the early phase of ripening when exposed to 4% NaCl alone (Figure 3). Meanwhile the 2.8% NaCl treatment alone included in the beef-based model did not show any improved *Listeria* inhibition indicating that the improved effect observed for the NaCl and KL combination was due to KL inclusion.

A comparison of growth kinetics conducted in MSB media using strain cocktails also showed similar behavior in the sensitivity of *L. innocua* and *L. monocytogenes* to the high NaCl salt concentration and the low NaCl-KL salt combination (Appendix A). Overall, the *L. innocua* strain cocktail displayed higher resistance compared to the examined *L. monocytogenes* genetic lineage I and II strains cocktails. Meanwhile within *L. monocytogenes*, the lineage II strain cocktail was more sensitive than the lineage I strain cocktail to both the low NaCl plus KL combination and the high salt (4% NaCl) concentration. This observation is in agreement with previous studies showing increased osmotic stress sensitivity of *L. monocytogenes* lineage II strains than those of lineage I [58,73,74]. 

In contrast with other reports [12], the treatments as applied in our experimental set up seemed bactericidal at some time points (Figure 3 and Figure 4 and Table 4). We presume this effect could have arisen from the synergistic effects of the NaCl, KL, pH, starter culture and other intrinsic or extrinsic food or media matrices factors. Previous studies have also demonstrated that under certain conditions exposure to sodium lactate (6%) alone can also be bactericidal [75,76]. We, however, cannot rule out if the formation of nonculturable but viable cells could also be responsible for this seemingly bactericidal effect. 

### 3.4. The Low NaCl (2.8%) Plus KL (1.6%) Combination Shows Similar Anti Listeria Activity as the High Salt (4% NaCl) Concentration Treatment during Salami Production and Storage

Studies of *L. monocytogenes* performed in the laboratory and those designed to mimic foods or food-processing systems can be limited in their ability to replicate food or environmental conditions since some food matrices intrinsic and extrinsic factors cannot be recreated in a test tube [77]. As such we also assessed the behavior of *Listeria* exposed to high NaCl salt and the low NaCl plus KL combination during production and storage of actual salami artificially inoculated with *Listeria.* The salami challenge test setup simulated salami stored whole which does not undergo further processing post ripening. In this setup *Listeria* introduction is most likely but not exclusively at the ingredients mixing and sausage production stage through contaminated raw materials, equipment, or personnel. There was no *Listeria* spp. detected on the un-inoculated negative control salami (batch 1), indicating that the salami sausages subsequently inoculated with the *L. innocua* strain cocktail in our study were free of detectable natural *Listeria* contamination. During the ripening of salami treated with varying NaCl and KL concentrations and combinations (Table 2) the growth behavior of the *L. innocua* strain cocktail although variable followed a similar overall profile (Figure 4). During the first 4 to 5 days of ripening KL-treated salami batches (batch 4: 2.8% NaCl + 1.6% KL, and batch 5: 2% NaCl + 2.6% KL) showed lower *L. innocua* CFU counts and a higher fold decrease in growth than NaCl-only treated batches. The NaCl and KL replacer combinations exhibited better anti-*Listeria* activity at this stage of ripening than the reduced salt (batch 3: 2.8% NaCl) and standard high salt (batch 2: 4% NaCl) alone preparations. Notably, the *L. innocua* salami growth profiles mirrored those observed during salami ripening simulation trials in MSB and beef (Figure 3). Growth profiles and corresponding growth factor changes showing the greatest CFU count decrease corresponded to the period when the pH dropped the most (days 2 to 4) (Figure 4 and Figure 5). This highlights the importance of a fast pH drop to the antimicrobial effect of salami recipe. However, this drop should not be too fast as the isoelectric point of some enzymes is reached too early, thereby reducing their activity. Inadvertently, this can lead to rancid, softer, and browner than red salami with a sour and tangy taste.

From day 7 to 20 the standard high NaCl salt (4% NaCl) concentration treated salami displayed better anti-*Listeria* activity as it maintained lower bacterial counts while growth occurred to different levels on the salami treated with the two NaCl plus KL combinations (Appendix A). As the pH increased from day 7 to day 16 there was bacterial growth at varying levels albeit greater in the salami treated with 2.8% NaCl and the low NaCl plus KL combinations (Figure 4 and Figure 5). The pH profile of the KL-treated salami which shows a reduction in pH drop and faster rise which could not have been predicted from simulation trials, represents a spoilage risk in fermented products. However, this may be counteracted, at least in part, by the antimicrobial activity of the lactate ions [78,79]. 

Contamination of RTE foods can occur during handling and packaging [5,26]. Evaluation of growth profiles of the bacteria during storage gives insights on the antimicrobial activity of the salami recipe if they were to be contaminated before storage. At seven days of storage at 4 °C (day 27), the standard high salt (4% NaCl) concentration treated salami showed *Listeria* growth whereas the low salt alone (2.8% NaCl) had the highest bacteria counts of all the salami batches. The low NaCl plus KL combination treated salami batches (2.8% NaCl + 1.6% KL and 2% NaCl + 2.6% KL) on the other hand both displayed drops in bacterial numbers suggesting that prolonged storage of ripened NaCl plus KL combination treated salami might further improve its public health safety. We confirmed that indeed the KL-treated salami continued to lower the *L. innocua* numbers during long-term storage (Table 4 and Figure 4). This will not only improve the safety of the salami but might further prolong its shelf life beyond that of 4% NaCl-only treated salami. Similarly, previous work has demonstrated that a KL and sodium diacetate (SD) combination have enhanced inhibition of *Listeria* growth on RTE meats during long-term refrigerated storage than when either compound is used alone [45,48].

We also observed that cold stress exposure of *Listeria* spp. in the 4% NaCl salami preparation during the first week of storage resulted in a slight spike in growth suggesting preadaptation of *Listeria* to cold stress probably because of osmotic stress preexposure during salami ripening. Such salt and cold stress cross-protection of *Listeria* is further justification for reducing salt levels in such processed foods, as this spike was not observed in the low salt-treated salami batches. Meanwhile, NaCl inclusion levels of 2.8% had the least inhibitory effect on growth potential (Table 4). The increased growth of *L. innocua* in the 2.8% NaCl-alone treated salami (30% salt reduction) during ripening and long-term storage, stresses the need for salt replacers that can maintain antimicrobial activity when salt reduction is done in processed foods. Overall, the final CFU counts at the end of ripening and even during storage were lower than the day zero starting inoculum counts (Table 4). This shows that all the salami batches maintained inhibitory activity against *Listeria* with an overall bactericidal effect as well. 

*L. monocytogenes* exposure to osmotic, pH, and low temperature stresses associated with salami ripening and storage can cause cell separation defects during cell division resulting in cell chaining. This potentially causes an underestimation of viable bacteria numbers on plate counts since a chain of several cells will only produce a single colony [80,81]. In our study, there were no significant differences in bacterial chaining observed among the different treatments. Therefore, it is highly unlikely that differences observed in CFU counts could have been distorted because of the differences in bacterial chaining under the different salami treatment conditions. It is also important to point out that osmotic and pH stress in food is postulated to increase the virulence of *L. monocytogenes* by priming the bacteria to digestive stress [21,22,82], hence it would be prudent in future studies to determine the consequences of such KL-based intervention on the virulence of *L. monocytogenes*. 

The *Listeria* contamination levels applied in our experiments were higher compared to those expected in natural contaminations [26,83,84,85]. Such high bacteria concentrations were however necessary in order to allow easy detection and quantification (log CFU reduction) of the effects of the different salting interventions on the fate of contaminating *Listeria* within our experimental setup. Moreover, the high starting *Listeria* concentrations were necessary in order to limit the impact of the Jameson effect. The high concentrations of the salami starter culture would create an additional impediment to the growth of organisms present in lower numbers [12,86]. 

Low number of *L. monocytogenes* are typically present in processed meats during manufacturing. The delay in growth induced by the presence of NaCl and KL in combination with the growth suppressing effects of the starter culture or other commensal microbiota, might be expected to further reduce the risk of listeriosis from processed meats such as salami. It is plausible that by the time contaminating *Listeria* are able to grow, the other bacteria present might have reached concentrations that are high enough to suppress *L. monocytogenes* growth. The magnitude of this suppression depends on the relative concentrations of starter culture and *L. monocytogenes* initially present. For these reasons, it is important that our findings are verified in future studies using salami preparations inoculated with lower *L. monocytogenes* numbers resembling those commonly found on salami and other processed meats. However, previous work in fermented sausages using low initial *L. monocytogenes* inoculum (<1.0 log CFU/g) showed that KL had a similar antimicrobial effect as the standard NaCl [78].

### 3.5. Water Activity and pH in KL-Treated Salami 

Previous work has suggested that simple salt reduction in fermented products cannot be made because of the low water activity that has to be reached in order to control the microbiota [13,44]. As a result, the technological and microbial safety as well as the sensory properties of the substitutes compared to NaCl will determine the extent of NaCl reduction.

In the salami ripening trial, significant difference in water activity was only noted in the 2.8% NaCl-only treated salami batch (Table 5), in fact, the KL-treated salamis had slightly lower water activity values among the *Listeria*-inoculated salami. Temperature, pH, and water activity are important factors influencing the ability of *L. monocytogenes* to grow in food. Foods with water activity and pH less than or equal to 0.920 and 4.4, respectively, or a combination of pH ≤ 5.0 and a water activity ≤ 0.940 are postulated not to support *L. monocytogenes* growth [63]. This was corroborated in part in our study as the water activity of the KL-treated salami was ≤0.908 and no growth was observed. However, on the non-KL but NaCl-treated salami (batches 2 and 3) having water activity in the range of 0.906 to 0.916 which is below the proposed 0.920, minor growth was observed within the first seven days of storage (0.18 log CFU) but not during long-term storage for the standard salt (4% NaCl) treated salami (Figure 4). This could be due to uneven distribution of the bacteria or salt such that in some areas of the salami the water activity was above 0.920 or had higher bacteria counts which allowed bacterial growth or give a false impression of growth. Also, this could be related to the adaptation of the bacteria to osmotic stress exposure during the ripening period, for example, previous studies have demonstrated that preexposure to osmotic stress can induce cross-protection or increased tolerance to other stressors such as bile, cold, pH, nisin, and hydrogen peroxide stress [14,15,23,24]. Alternatively, the strains used in our study might have evolved to grow at this water activity. Suggesting a need to reevaluate this 0.92 water activity value taking into account the differences in food matrices and bacterial strains. However, it is important to note that this minor growth is significantly lower than the cutoff of 0.5 log CFU/g stipulated as the defining point above which a food product is said to support the growth of *Listeria* spp. [87,88]. This is determined after challenge tests, many of which include exposure of the food product to temperatures that are higher than the storage temperature [88]. This allows for faster growth if the product supports *Listeria* growth, however, this was not done in our study. A limitation of this study might be that for the storage phase of the challenge test, the salami sausages were only stored at 4 °C and not exposed to abusive temperatures. This limits flexibility in the interpretation of the results as they might only be valid for the product stored under these specific temperature conditions. Further experiments will have to be performed under abusive temperature conditions that the salami might be exposed to during transportation, retail, and at the consumer.

Low pH inhibits the growth or toxin production of microbes such as *Staphylococcus aureus*. It is critical that pH below 5.3 are achieved within the stipulated degree-hours for the product depending on the fermentation conditions. All the salami batches reached pH below 5.3 within 48 h, albeit at a faster rate in the NaCl only treated salami (Figure 5, Appendix A). They held this pH until day 7 to 14 depending on the salt treatment. From there the salami pH increased to >7 at the end of ripening and during storage. Therefore, pH is less likely to have had a contribution to the growth inhibition observed during storage. This pH increase is in part due to proteolytic activity as well as the metabolic action of the *P. nalgiovensis* molds and *S. xylosus* used in the ripening process. The pH rise was faster in KL-treated salami probably as a result of the buffering activity of the lactate [70,89]. 

Taken together our findings show that predictive models though very useful have limitations in evaluating the capacity of foods to support *L. monocytogenes* growth. For example, the simulation trials could not predict these growth or pH profile changes (Figure 4 and Figure 5), in part because molds could not be used in the MSB or beef experimental setup. The results, moreover, highlight the importance of validating strategies for the control of *L. monocytogenes* on the actual foods they are intended for.

### 3.6. Effect of the Different Salt Treatments on the Starter Culture Bacteria 

The beneficial effects of any salt reduction intervention in reducing risk from *L. monocytogenes* could be diminished if the growth suppression is applied equally to the starter culture. As such, we also assessed the starter culture growth profiles under the different salami salt inclusion levels. This showed that starter culture growth profiles remained comparable between the standard high salt (4% NaCl) alone and the reduced NaCl and KL combinations (Figure 6). These observations are in agreement with previous work that reported a limited effect on microbiological stability upon partial NaCl substitution with KL [78]. 

### 3.7. Salami Quality Is Maintained with the Different Salt Treatments 

Sodium chloride functions to improve color, flavor, texture, water-holding capacity, fat binding, and prolongs shelf-life in meat products. Potassium lactate shares most of these properties. In addition, lactates function as flavoring agents, color stabilizers, and can be useful for their effects on retardation of lipid oxidation and subsequent off-odor development [68,70,90,91]. Salami treated with different NaCl salt concentrations as well as combinations of NaCl and KL all displayed similar color, texture, and firmness (Appendix A). As water activity, pH, weight loss and starter culture growth profiles were generally similar between the salami batches it is highly likely that there will be minimal differences in the quality of the salami (Figure 5 and Figure 6, Table 5). To confirm this, we evaluated the final salt content, fat, protein, carbohydrate, and ash content as well as Warner–Bratzler measurements of the different salami preparations. This allowed for an objective assessment of the quality of the salami. Results from proximate composition analyses revealed that there were no major differences in the levels of dry matter, carbohydrates, fat, moisture, and protein among the salami batches (Table 6). As expected, however, the analyses revealed noticeable differences in sodium, ash, and salt levels among salami formulated with low (2 or 2.8%), and high (4%) levels of NaCl (Table 6). 

The maximum possible NaCl substitution rates range from 25 to 40%, which is dependent on the product in question with the aim to maintain product quality, flavor, and consumer acceptability [43,78,92]. In fermented sausages and dry cured loins, it was possible to substitute 40% of NaCl with potassium lactate without any significant detrimental effect on flavor. Above this inclusion level, a slight potassium lactate or an undesirable sweet taste was detected [92]. This would be expected in the 2% NaCl plus 2.6% KL-treated salami where NaCl reduction was 50%. At this high NaCl reduction level the salami quality and texture might change to a level detectable by the consumer. In our study, the salami with 50% reduction though physically similar to the other salami might be softer than the other salami batches. Based on the Warner–Bratzler measurements the salami batch with 50% NaCl reduction (2% NaCl + 2.6% KL) was deemed softer as demonstrated by the reduced maximum force or energy required to cut through it in comparison to salami from the other batches (Table 6; Appendix A). The 30% NaCl reduction employed in the 2.8% NaCl plus 1.6% KL-treated salami falls within the recommended range where no significant differences in flavor and market acceptance are expected. Furthermore, previous work has also shown that KL (1.36% to 2.1%) can be added to RTE meats such as cold smoked salmon with no observable flavor or proximate composition differences [9,48]. Combining these observations by others with previous organoleptic analysis results of salami prepared with this recipe (Agroscope Switzerland, unpublished) suggests that KL can be a suitable NaCl replacer that maintains quality and antimicrobial food safety standards of salami. However, a potassium lactate flavor and a reduction in saltiness, as well as texture change, might be detected by some consumers at these levels of replacement. With this in mind, changes to the formula must be introduced gradually to ensure that consumers adapt to the new taste. The use of KL as a NaCl replacer can be augmented by combining such an intervention with the use of flavor enhancers such as L-Lysine which enhances the saltiness of products. Moreover, optimizing the physical form of salt so that it becomes more functional and taste bioavailable might improve saltiness and assist in maintaining customer acceptability.

### 3.8. Overall Benefits and Potential Drawbacks of KL Use as a NaCl Replacer 

Meat matrices factors, storage temperature, and packaging conditions may affect the effectiveness of lactates [46,48,67,78,92,93], hence further analysis needs to be done to verify if the texture, flavor, and antimicrobial effect of the NaCl-KL combination varies from one product to another. It has been proposed that commercial usage levels of KL (2.0%) and sodium diacetate (0.15%) alone are not sufficient to control *L. monocytogenes* in case of pathogen contamination [9,46]. The combination of KL and NaCl as used in our study can be an alternative to ensure efficacy for microbial growth inhibition by KL as well as negating the salt stress tolerance of *L. monocytogenes*. Further efficiency can be achieved by combining this technology with other hurdle techniques such as vacuum packaging, modified atmosphere packaging, and high-pressure treatment [9]. Future challenge test studies using salami prepared with this NaCl plus KL formula being inoculated with cold adapted *Listeria* post ripening are required to further validate our findings on inhibition of bacteria introduced post ripening which might happen during slicing and packaging. In addition, studies are needed to verify the effects of this KL based intervention on growth kinetics of other *L. monocytogenes* genetic backgrounds as well as its efficacy on other pathogens associated with salami.

One of the potential barriers to salt replacement and acceptability of a replacer by meat processors is cost as salt is one of the cheapest food ingredients available. Fortunately, potassium lactate is also a relatively cheap ingredient, therefore, cost is not expected to be a hindrance to its use as a salt replacer. Moreover, in fermented meats, the addition of lactate has the extra advantage of retarding toxin production by *Clostridium botulinum* which is a major public health threat in such products [90]. Unlike KL, NaCl alone is not able to retard this toxin production.

When making such substitutions the possible negative health effects on some subgroups within the population need to be emphasized. High potassium foods could have negative impacts on individuals with ailments including Type I diabetes, chronic renal insufficiency, end-stage renal disease, severe heart failure, and adrenal insufficiency [13,94]. With that said it has been shown that a potassium-rich diet reduces the negative effects of sodium on blood pressure with a maximum daily intake recommendation of 4.7 g potassium/day [94]. 

## 4. Conclusions

The reduction of added NaCl salt in foods including fermented meat products has been proposed to decrease the amount of sodium in the diet as this has associated health benefits. We have shown here that KL is an effective salt replacer allowing lower NaCl levels (2.8% vs. 4%) to be applied without compromising product quality and antimicrobial benefits associated with the higher NaCl salt concentration currently applied in standard salami production. Combining such an intervention with improved manufacturing practices and other hurdle techniques will increase food safety and potentially extend product shelf life. Overall, this intervention contributes to the control of pathogens like *L. monocytogenes* that continue to be public health threats to consumers. We also show that MSB-based models can be applied for preliminary screening of antimicrobial effects on pathogens such as *L. monocytogenes* and thus might be used to test alternative ingredients for effects on starter culture and pathogen growth. 

## Figures and Tables

**Figure 1 foods-10-00114-f001:**
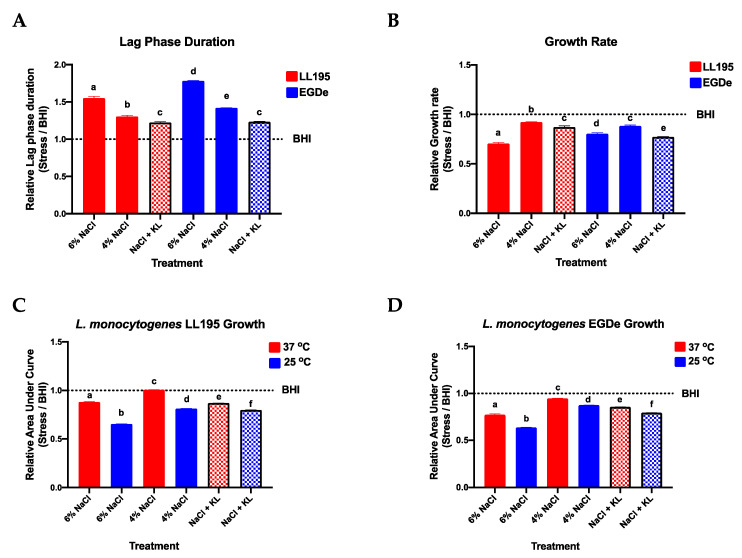
The low NaCl (2.8%) plus KL (1.6%) combination retains antimicrobial activity against *L. monocytogenes* (LL195 and EGDe) and temperature has a synergistic effect on the antimicrobial activity of both high NaCl (4% and 6%) concentration alone and the low NaCl plus KL combination (NaCl + KL). (**A**,**B**) At 37 °C *L. monocytogenes* lag phase durations were increased while growth rates were reduced through both interventions. (**C**,**D**) The application of high NaCl concentrations alone (4% and 6% NaCl) and the low NaCl plus KL combination (bars with patterns) both produced greater overall *Listeria* growth inhibition at lower (25 °C vs. 37 °C) temperature. Bars and error bars represent relative mean (**A**) lag phase duration (**B**) growth rate and (**C**,**D**) area under the curve generated using *opm* and GraphPad Prism from three replicates observed from kinetic growth assays in BHI supplemented with NaCl and KL at various concentrations and combinations. Each growth parameter per treatment presented are means expressed relative to those of the control with no stressor added. The control is represented by the dotted line labeled BHI. Different letters indicate a significant difference between strains (*p* < 0.05 based on one-way ANOVA and Tukey post-hoc test pairwise comparison).

**Figure 2 foods-10-00114-f002:**
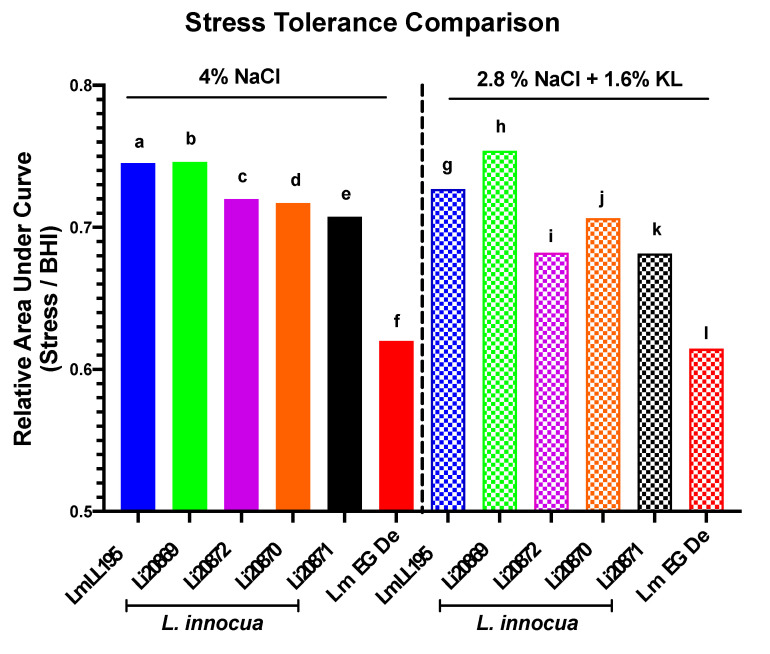
The selected *L. innocua* surrogate strains display similar stress response behavior to *L. monocytogenes* during growth in BHI supplemented with 4% NaCl or the 2.8% NaCl plus 1.6% KL combination. GraphPad Prism generated data represent relative area under the curve (AUC) derived from growth curves of *L. monocytogenes* (LL195 and EGDe) and *L. innocua* (*Li*20869, *Li*20870, *Li*20871, and *Li*20872) strains grown in normal BHI vs. growth under stress (BHI supplemented with 4% NaCl or 2.8% NaCl plus 1.6% KL). Each mean AUC per strain and treatment was expressed relative to those of the control with no stressor added. Based on the AUC, the selected *L. innocua* strains distributed within the range of low (EGDe) to high (LL195) osmotolerant *L. monocytogenes* strains. Different letters indicate a significant difference between the strains and treatments (*p* < 0.05 based on one-way ANOVA and Tukey post-hoc test pairwise comparison).

**Figure 3 foods-10-00114-f003:**
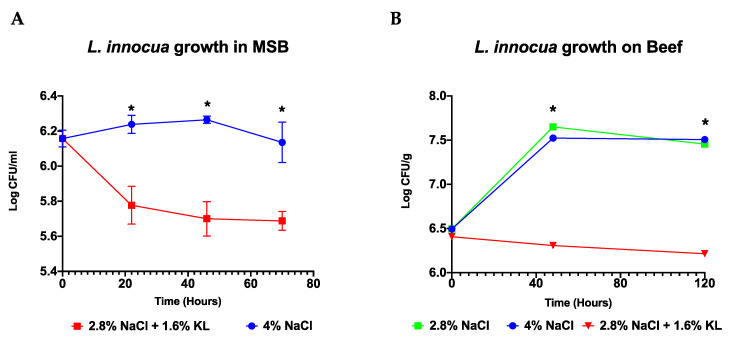
Growth behavior of a four strain *L. innocua* cocktail under simulated salami ripening conditions in the (**A**) MSB and (**B**) on beef-based models. The low NaCl (2.8%) plus KL (1.6%) combination displayed superior anti *Listeria* activity compared to the traditional salami recipe of high salt (4% NaCl) concentration. Data are mean log CFU counts from three biological replicates. * Indicates significant difference between treatments at that time point (*p* < 0.05 based on one-way ANOVA and Tukey post-hoc test pairwise comparison).

**Figure 4 foods-10-00114-f004:**
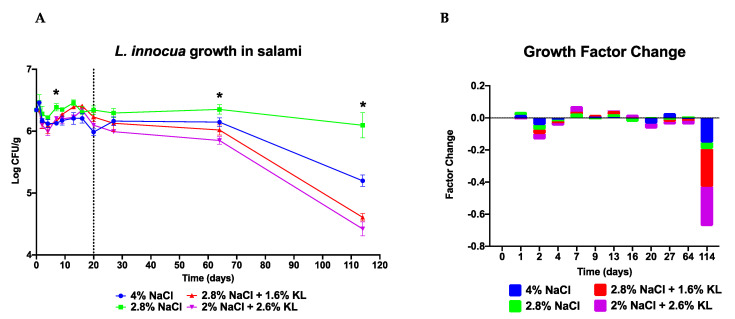
*L. innocua* growth during ripening and storage of artificially contaminated salami produced with high salt (4% NaCl) and low salt (2.8% NaCl) concentrations alone as well as the two different low NaCl plus KL combinations (2.8% NaCl + 1.6% KL or 2% NaCl + 2.6% KL). Presented is (**A**) growth profiles and (**B**) relative growth increase or decrease quantification of the four *L. innocua* strain cocktail under the different treatments. Data are mean log CFU counts from two replicates at 0, 1, 2, 4, 7, 9, 13, 16, and 20 days of salami ripening. Day 27, 64 and 114 represent 7, 44 and 94 days of storage at 4 °C. The dotted line denotes the end of salami ripening. * Indicates significant difference between treatments at that time point (*p* < 0.05 based on one-way ANOVA and Tukey post-hoc test pairwise comparison of the salami batches).

**Figure 5 foods-10-00114-f005:**
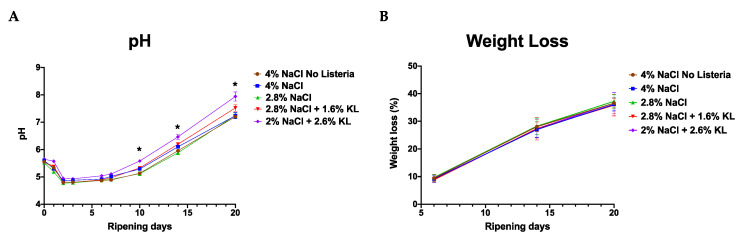
The (**A**) pH and (**B**) weight loss profiles during ripening of salami batches prepared with different NaCl and KL levels and combinations. Both pH and weight loss profiles were comparable between the different salami treatments albeit higher in the KL-treated salami at some stages. Data are mean pH (*n* = 6 per batch) and weight loss readings (*n* = 7 per batch) from two separate salami ripening trial repetitions observed during 20 days of salami ripening. * Indicates significant difference between salami batches (*p* < 0.05 based on one-way ANOVA and Tukey post-hoc test pairwise comparison of the salami batches).

**Figure 6 foods-10-00114-f006:**
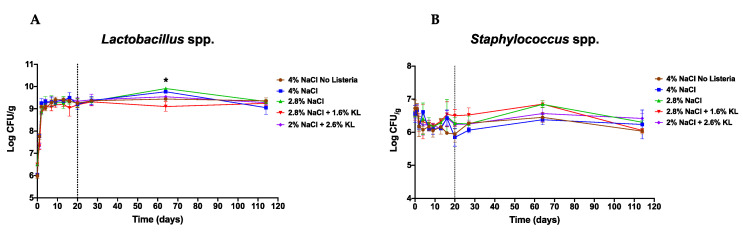
Growth profiles of (**A**) *Lactobacillus plantarum* and (**B**) *Staphylococcus xylosus* during salami ripening and storage. The starter culture bacterial growth profiles were comparable between the different salami batches. Shown are mean log CFU numbers from two experimental repeats observed on 0, 1, 2, 4, 7, 9, 13, 16, and 20 days of salami ripening. Day 27, 64, and 114 represent 7, 44, and 94 days of storage at 4 °C. The dotted line denotes the end of salami ripening. * Indicates significant difference between salami batches (*p* < 0.05 based on one-way ANOVA and Tukey post-hoc test pairwise comparison of the salami batches).

**Table 1 foods-10-00114-t001:** Strains used in this study.

Strain ID	Description	References
*L. innocua* (*Li*)		
*Li*20869	Agroscope reference strain (Surface of a semi-hard smear cheese, 2005)	Agroscope Switzerland
*Li*20870	Agroscope reference strain (Surface of a mountain cheese, 2005)	Agroscope Switzerland
*Li*20871	Agroscope reference strain (rind of a swiss cheese, 2005)	Agroscope Switzerland
*Li*20872	Agroscope reference strain (cheese smear, 2006)	Agroscope Switzerland
*L. monocytogenes* (*Lm*)		
*Lm* N2306	2013–2014 Swiss listeriosis outbreak, lineage I, serotype 4b, CC4	[49]
*Lm* N1546	2011 Swiss listeriosis outbreak, lineage II, serotype 1/2a, CC8	[50]
*Lm* Lm3136	2005 Swiss listeriosis outbreak, lineage II, serotype 1/2a, CC18	[51]
*Lm* Lm3163	2005 Swiss listeriosis outbreak, lineage II, serotype 1/2a, CC26	[51]
*Lm* LL195	1983–1987 Swiss listeriosis outbreak, lineage I, serotype 4b, CC1	[52]
*Lm* N16-0044	2016 Swiss listeriosis outbreak, lineage I, serotype 4b, CC6	[53]
*Lm* EGDe	Rabbit isolate and reference strain, lineage II, serotype 1/2a, CC9	[54]

**Table 2 foods-10-00114-t002:** Salami preparation and inoculation strategy.

Batch Number	Name	NaCl [g/kg]	*L. innocua* Addition	Potassium Lactate (KL)	Final Salt Content after Ripening
1	Reference salt (4% NaCl no *Listeria*)	26 g/kg	no	no	4% NaCl
2	Reference salt (4% NaCl)	26 g/kg	yes	no	4% NaCl
3	Reduced Salt (2.8% NaCl)	18 g/kg	yes	no	2.8% NaCl
4	Reduced Salt (2.8% NaCl + 1.6% KL)	18 g/kg	yes	16 g/kg	2.8% NaCl + 1.6% KL
5	Reduced Salt (2% NaCl + 2.6% KL)	13 g/kg	yes	26 g/kg	2% NaCl + 2.6% KL

**Table 3 foods-10-00114-t003:** Comparison of *L. monocytogenes* EGDe and LL195 growth rates under osmotic stress ^a^.

Condition	LL195 Growth Rate µ_max_ (OD_600_/hr)	EGDe Growth Rate µ_max_ (OD_600_/hr)
BHI	0.184 ± 0.001	0.149 ± 0.001
4% NaCl	0.170 ± 0.001	0.131 ± 0.001
6% NaCl	0.130 ± 0.002	0.119 ± 0.002
2.8% NaCl + KL 1.6%	0.161 ± 0.001	0.115 ± 0.000

^a^ Presented values are mean growth rates and their standard deviations generated using *opm* from three biological replicates observed from kinetic growth assays (24 h, 37 °C) in normal (BHI only) as well as BHI supplemented with 4% and 6% NaCl as well as the NaCl (2.8%) plus KL (1.6%) combination.

**Table 4 foods-10-00114-t004:** Comparison of *Listeria* growth potential in the different salami batches ^a^.

Time (Days) ^b^	4% NaCl	2.8% NaCl	2.8% NaCl + 1.6% KL	2% NaCl + 2.6% KL
20	−0.358	−0.002	−0.113	−0.246
27	−0.179	−0.048	−0.216	−0.350
64	−0.196	0.009	−0.323	−0.492
144	−1.144	−0.247	−1.734	−1.922

^a^ Presented values are calculated as log CFU/g for that day minus day zero inoculum log CFU/g counts. Negative values correspond to lower log CFU count in comparison to day zero inoculum meaning that *Listeria* failed to grow, and its numbers were reduced. ^b^ Day 20 represents end of ripening whilst day 27, 64 and 144 represent 7, 44, and 94 days of storage at 4 °C, respectively.

**Table 5 foods-10-00114-t005:** Water activity.

Batch Number	Name	NaCl Change	Day 20 (a_w_) *	Day 114 (a_w_) *
1	4% NaCl no *Listeria*	Reference salt	0.890 ± 0.004	0.894 ± 0.005
2	4% NaCl	Reference salt	0.906 ± 0.010	0.908 ± 0.002
3	2.8% NaCl	30% Reduction	0.912 ± 0.001	0.916 ± 0.005
4	2.8% NaCl + 1.6% KL	30% Reduction	0.903 ± 0.013	0.892 ± 0.015
5	2% NaCl + 2.6% KL	50% Reduction	0.900 ± 0.006	0.908 ± 0.010

* Data are mean water activity (a_w_) and standard deviation values determined on day 20 (end of ripening) and day 114 (94 days of storage at 4 °C).

**Table 6 foods-10-00114-t006:** Proximate composition and Warner Bratzler measurements.

Parameter (per 100 g)	4% NaCl No *Listeria*	4% NaCl	2.8% NaCl	2.8% NaCl + 1.6% KL	2% NaCl + 2.6% KL	Method
Drying loss (Water content)	37.7 g	37.9 g	35.9 g	34.5 g	36.7 g	LCAMET33 Gravimetric
Dry matter	62.3 g	62.1 g	64.1 g	65.5 g	63.3 g	LCAMET33 Gravimetric
Ash ^a^	5.5 g	5.41 g	4.21 g	5.16 g	4.41 g	LCAMET33 Gravimetric
Total fat	32.7 g	32.1 g	33.2 g	33.7 g	33.1 g	LCAMET36 Gravimetric
Total protein (f = 6.25)	26.9 g	26.3 g	28.4 g	26.9 g	25.9 g	LCAMET34 Dumas
Sodium (mg/Kg) ^a^ CAS 7440-23-5	17,000 mg	17,000 mg	13,000 mg	13,000 mg	9200 mg	LCAMET05 IC
Calculated Values						
Energy (kJ)	1680 kJ	1626 kJ	1697 kJ	1717 kJ	1663 kJ	SQTSMET02
Energy (kcal)	405 kcal	392 kcal	409 kcal	414 kcal	401 kcal	SQTSMET02
Total carbohydrates	<0.5 g	<0.5 g	<0.5 g	<0.5 g	<0.5 g	SQTSMET02
Salt (sodium × 2.5) ^a^	4.25 g	4.25 g	3.25 g	3.25 g	2.3 g	SQTSMET02
Warner–Bratzler measurements						
Force maximum (N)	112 ± 4	98 ± 19	121 ± 17	121 ± 8	83 ± 6 *	

^a^ Lower ash, sodium and salt levels observed in salami prepared with reduced salt recipes. * Indicates significant difference between salami batches. N: Newtons.

## Data Availability

The data presented in this study are available in the article and Appendix A (https://www.mdpi.com/2304-8158/10/1/114/s1).

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
