# Peer review of "Potassium Lactate as a Strategy for Sodium Content Reduction without Compromising Salt-Associated Antimicrobial Activity in Salami"

_foods, 2021, doi:10.3390/foods10010114_

Round 1
Reviewer 1 Report
General comments
The topic is interesting, there is not so much in literature. In particular there are no references on the effects of this specific association (KL – NaCl) on Lm, that could be particularly interesting because NaCl is normally present as ingredient in many meat products, and there is interest to reduce it for health reasons.
The paper is well introduced and the references are fine. Some more references on studies on the effects of Potassium lactate-sodium diacetate could be added (e.g. Jin et al. 2005 “Effects of temperature, pH, and potassium lactate on growth of Listeria monocytogenes in broth” ; Vogel et al. 2006 “Potassium Lactate Combined with Sodium Diacetate Can Inhibit Growth of Listeria monocytogenes in Vacuum-Packed Cold-Smoked Salmon and Has No Adverse Sensory Effects”; Abou-Zeid et al. 2007 “Survival and Growth of Listeria monocytogenes in Broth as a Function of Temperature, pH, and Potassium Lactate and Sodium Diacetate Concentrations).
My main concerns are about how materials and methods are described and, consequently, about how results are displayed. Many details are missing and should be specified.
The authors do not refer neither to the EURL Listeria monocytogenes technical guidance document for conducting shelf life studies, nor to the ISO 20976-1:2019, therefore they should always give detailed information about how they carry out inoculum preparation, inoculation, calculation of the growth rate, and justify the reliability of the alternative methods that they used.
In particular I have major concerns regarding the methods used for the enumeration of Listeria spp. and Listeria monocytogenes in BHI supplemented with NaCl or KL. Why did the authors not use plate count? OD is not as accurate as plate count. At least some reference to other studies or validation references should be provided, in order to evaluate if OD measurement could be a reliable method in this case.
I have concerns also regarding the calculation of the growth rate in BHI supplemented with NaCl or KL. It is not clear how the authors converted OD values to Log CFU/g values: actually to measure the growth rate with DMFit (ComBase) it is necessary to give Log CFU/g values at different times. The authors only give relative means of the growth kinetics parameters (lag phase and growth rate) comparing “stress” to “BHI”. That gives the results in a glance, but more details should also be given in the main manuscript, not only in the supplementary material. I see in the supplementary material that growth rate is expressed in OD600/hour, but DMFit only calculates growth rate values expressed in Log CFU/g per hour or day.
Also the terms used relating to the growth rate are not uniform: in certain cases the authors use “growth potential”, in other “growth rate”, in other “maximal growth rate”, in other “maximum growth rate”.
Specific comments
Line 112: I think it is “garlic”, not “galic”
Line 129: it is not “growth potential” but “growth rate”. Please refer to official documents such as the
Lines 137-140: I don’t feel familiar with the use of OD, I would prefer ISO plate count. However it should be at least detailed how the OD was converted into log CFU/g, that is the unit used in DMFit to calculate the growth rate. I don’t think that it could be possible to calculate a growth rate with DMFit using OD600/hours as unit. Listeria monocytogenes enumeration with OD should be an internal method, therefore some sort of validation in comparison with ISO or at least literature reference is needed. Please give also some literature reference also for the use of the AUC for bacterial growth kinetics description (I am not familiar with that, but as far as I know it is mostly used for pharmacokinetics)
Line 138: perhaps the authors mean “maximum growth rate”. Actually the right term would be “growth rate” or “maximum rate” , if the authors actually used DMFit to calculate it. In fact, DMFit accepts data expressed in decimal logarithm CFU/g (log CFU/g), and gives back results expressed in log CFU/g per time unit, that could be hour or day (“maximum rate”). The maximum growth rate (µmax) is instead usually expressed in natural logarithm, and it is calculated in this unit by other software such as Symprevius. This is detailed also in the EURL Lm Technical Guidance Document for conducting shelf life studies, to which the author refer only in discussion.
Line 140: DMFit is available from www.combase.cc, this should be reported in the references, other than the citation of the Baranyi & Roberts paper.
Lines 158-166: the use of the ISO 11290-2:2017 would be more opportune. However it is OK, as also according to ISO the PBS could be used in place of peptone water.
Line 211: it is necessary to precisely specify at what times water activity and pH were determined. What do the author mean with “during storage”? Is it ripening? Where are the results displayed? I can see in figure 5 the pH values at different ripening times, but I cannot find aw values. It would be opportune to calculate water activity at least at the beginning of the ripening, to have an idea of the water activity during the whole repining period, as it is a crucial factor to understand if the product supports or not the growth of Listeria.
Lines 247-248 and figure 1: growth rates results should be detailed. In the text the authors write about “resulting in growth rate reduction levels” and refer to the figure 1, but in the figure only “means expressed relative to those of the control with no stressors added” are displayed. The actual growth rate results, with the unit used e.g. log CFU/g/ hour, should be given in detail in the main manuscript.
Line 293 (figure 2): there is a mistype in figure 2 (innocua with the double c)
Lines 314-329 and figure 3-4: only diagrams are displayed to compare L. innocua growth kinetics in MSB and Beef in the different combinations of NaCl alone and NaCl+KL. It is the same for salami in figure 4. In the supplementary material I see tables S5 displaying the “comparison of growth relative to day zero inoculum”. I understand that the calculation of growth rate could be not opportune considering that cocktails of strains were used. However it would be much more easier to have a simple table showing growth potentials (e.g. difference in log CFU/g between the end and the beginning of the experiment), to compare the effect of the use of the different salt concentrations.
Lines 452-454: when does the water activity of the KL treated salami was <0.908? At the end of ripening? And what about during ripening? Evolution of water activity during ripening should be clearly shown.
Line 470 (table 3): it is not clear to what moment of ripening the water activity value per each batch is referred. Why just one value per batch is displayed?
Reviewer 2 Report
The introduction is vey long
Methods are not well described
for example the composition of meat simulation borth was never specified
Strains of L. innocua were isolated and characterized in thsi study or in previous studies?
The experimental plant is not very clear
Reviewer 3 Report
The manuscript raises a very important issue from the point of view of the safety and health of consumers. It is written in a thoughtful and careful manner. Research methods are correctly selected and described. The results are well presented. The information contained in the manuscript has application potential.
However, there are a few minor drawbacks that authors should remove or complete:
- In the introduction, it is worth providing information on the percentage of meat samples in which monocytogenes is detected and information on larger epidemics related to the consumption of contaminated meat.
- Line 112: Garlic rather than galic
- How was the NaCl percentage by weight / volume determined?
- What was the number of bacteria in the suspensions determined, e.g. 10 ^ 5 etc.?
- Is the quantitative contribution of plantarum and S. xylosus in SA1 starter culture known?
Round 2
Reviewer 1 Report
The manuscript is now much improved. More details have been added where required. Method used are now well described. In particular it is now well clear how growth curves were calculated. Evaluation of water activity at different moments during ripening would have given even more interesting information relating to the possibility of Listeria monocytogenes to grow in the product in this stage. However there is a noticeable amount of interesting results also in the present form, giving useful information to the scientific community.
Reviewer 2 Report
The manuscrit was improved following reviewer's suggestion